# A Short and Unified Convergence Analysis of the SAG, SAGA, and IAG Algorithms

Feng Zhu [1]   Robert W. Heath Jr. [2]   Aritra Mitra [1]

## Abstract

Stochastic variance-reduced algorithms such as Stochastic Average Gradient (SAG) and SAGA, and their deterministic counterparts like the Incremental Aggregated Gradient (IAG) method, have been extensively studied in large-scale machine learning. Despite their popularity, existing analyses for these algorithms are disparate, relying on different proof techniques tailored to each method. Furthermore, the original proof of SAG is known to be notoriously involved, requiring computer-aided analysis. Focusing on finite-sum optimization with smooth and strongly convex objectives, our main contribution is to develop a single unified convergence analysis that applies to all three algorithms: SAG, SAGA, and IAG. Our analysis features two key steps: (i) establishing a bound on delays due to sub-sampling using simple concentration tools, and (ii) carefully designing a novel Lyapunov function that accounts for such delays. The resulting proof is short and modular, providing high-probability bounds for SAG and SAGA that can be seamlessly extended to non-convex objectives and Markov sampling. As an immediate byproduct of our new analysis technique, we obtain the best known rates for the IAG algorithm, significantly improving upon prior bounds.

## 1. Introduction

We consider the following finite-sum optimization problem:

$$\min_{x \in \mathbb{R}^d} f(x) := \frac{1}{N} \sum_{i=1}^{N} f_i(x), \tag{1}$$

where each $f_i : \mathbb{R}^d \to \mathbb{R}$ is assumed to be $L$-smooth, $f$ is $\mu$-strongly convex, and $x^* = \operatorname{argmin}_{x \in \mathbb{R}^d} f(x)$ is the mini-

mizer of the composite function $f(x)$. Problems of the form in (1) arise in the context of empirical risk minimization in machine learning, where $f(x)$ serves as a finite-sample approximation (based on $N$ samples) of a true risk function.

A natural way to solve (1) is to run the gradient descent (GD) algorithm. When $f$ is $L$-smooth and $\mu$-strongly convex, GD guarantees exponentially fast convergence to $x^*$, where the exponent of convergence depends on the *condition number $\kappa = L/\mu$* (Bubeck et al., 2015). This fast *linear* convergence rate, however, comes at the expense of $N$ gradient evaluations per iteration, which can be computationally demanding when $N$ is large. An appealing alternative is the stochastic gradient descent (SGD) algorithm (Robbins & Monro, 1951) which, at each iteration, moves along the negative gradient of just one component function chosen uniformly at random from the set $[N] := \{1, 2, \ldots, N\}$. While SGD evaluates only one gradient per iteration, the high variance in its update direction necessitates a diminishing step-size sequence to ensure exact convergence to $x^*$ with no residual bias. Unfortunately, this leads to a much slower sublinear rate (Moulines & Bach, 2011).

**Variance-Reduction Algorithms.** A breakthrough in this regard was achieved by Roux et al. (2012), who invented the stochastic average gradient (SAG) algorithm. Like SGD, SAG evaluates only a single gradient in each iteration, but exploits memory of past gradients of all components. Remarkably, this approach is able to retain the linear convergence rate of GD. However, the proof of this result in Schmidt et al. (2017) is notoriously challenging, and requires computer-aided analysis. Although a related algorithm called SAGA with a simpler proof was developed by Defazio et al. (2014a), the Lyapunov function in this paper fails to explain the convergence behavior of SAG. A deterministic variant of SAG, called the incremental aggregated gradient (IAG) algorithm, was developed by Blatt et al. (2007), and an explicit linear convergence rate for this algorithm was obtained in Gurbuzbalaban et al. (2017). However, the analysis of IAG, which is fundamentally different from those of SAG and SAGA, yields a much slower rate compared to SAG, SAGA, and GD.

Our main contribution is to develop a **single unified proof** that is surprisingly short and simple, and yields linear con-

---

[1]Department of Electrical and Computer Engineering, North Carolina State University, Raleigh, USA [2]Department of Electrical and Computer Engineering, University of California, San Diego, USA. Correspondence to: Aritra Mitra <amitra2@ncsu.edu>.

*Proceedings of the 43rd International Conference on Machine Learning*, Seoul, South Korea. PMLR 306, 2026. Copyright 2026 by the author(s).

vergence rates for SAG, SAGA, *and* IAG. Furthermore, our analysis significantly improves the best known rate for IAG.

In what follows, we elaborate on our main **contributions**, and discuss their implications in relation to prior work.

● **Unified Proof Technique.** Stochastic variance-reduced methods like SAG and SAGA, and their deterministic counterparts like IAG, all use memory of past gradients to maintain accurate estimates of the full gradient. However, despite the similarity in their update rules, existing convergence analyses of these algorithms differ considerably. In particular, the difficulty in analyzing SAG has often been attributed to the fact that its update direction is *biased*, unlike those for SGD and SAGA that admit relatively simpler proofs. In Gurbuzbalaban et al. (2017), the authors mention: "*We also note that most of the proofs and proof techniques used in the stochastic setting such as the fact that the expected gradient error is zero do not apply to the deterministic setting and this requires a new approach for analyzing IAG.*" Thus, whether a unified analysis framework can explain the dynamics of both biased and unbiased, stochastic and deterministic, variance-reduced algorithms is far from obvious. We show for the first time that this is indeed possible.

Our proof framework is simple, intuitive, and modular, and features two key steps. In the first step, for stochastic subsampling patterns, we use a concentration argument to control the maximum delay in seeing any component function. As a result, on a "good" event of sufficient measure, one can view both SAG and SAGA as delayed versions of GD, with an upper-bound on the delays that scales as $\tilde{\mathcal{O}}(N)$. Based on this insight, in the second step, we construct a novel Lyapunov function that maintains a window of stale gradients. The careful construction of this function is crucial to us achieving our desired rates. We then establish a one-step contractive recursion for this Lyapunov function, which translates to *high-probability* linear convergence rates for SAG and SAGA in Theorem 3.9. Our Lyapunov function and overall proof structure (outlined above) depart fundamentally from the analysis of SAG and SAGA.

● **High-Probability Bounds for SAG and SAGA.** The traditional analyses of stochastic optimization algorithms typically provide bounds that hold only in expectation. This is true for variance-reduced (VR) algorithms like SAG and SAGA as well, and the proofs of these algorithms in Roux et al. (2012); Schmidt et al. (2017); Defazio et al. (2014a) only provide in-expectation guarantees. Unfortunately, such guarantees only capture the "typical" behavior of the algorithm, and do not adequately represent rare/tail events. As a result, a series of high-probability bounds for SGD and its variants under different noise models have emerged over the last few years; see Li & Orabona (2020); Liu et al. (2023); Sadiev et al. (2023). Despite this rich literature, high-probability bounds for SAG and SAGA have remained

elusive. Theorem 3.9 closes this gap and contributes to a deeper understanding of these celebrated VR algorithms.

● **Bounds under Markov Sampling.** SAG and SAGA have been typically analyzed under I.I.D. sampling of component functions. We show in Theorem 3.12 that our analysis extends seamlessly to more general Markov sampling schemes. For this, we leverage a Bernstein concentration bound for uniformly ergodic Markov chains from Paulin (2015).

● **Improved Bounds for IAG.** The work of Blatt et al. (2007) that originally developed the IAG algorithm provided no explicit convergence rates for general smooth and strongly convex functions. This gap was later addressed by Gurbuzbalaban et al. (2017), who, for deterministic cyclic sampling patterns, established a convergence rate of $\mathcal{O}\left(\exp(-K/(\kappa^2 N^2))\right)$ after $K$ iterations of IAG. Here, recall that $\kappa$ is the condition number and $N$ is the number of component functions. Due to the quadratic dependence on $\kappa$ and $N$ in the exponent, the rate predicted for IAG in Gurbuzbalaban et al. (2017) is much slower compared to that for GD, SAG, and SAGA. Such a pessimistic picture for IAG has, in fact, prompted the development of more complex deterministic variance-reduction algorithms; see Mokhtari et al. (2018). As a byproduct of our analysis for SAG and SAGA, we establish a significantly tighter rate of $\mathcal{O}\left(\exp(-K/(\kappa N))\right)$ for IAG. Intuitively, this rate makes sense, since $N$ iterations of IAG are comparable to one iteration of GD. Thus, our unified analysis provides a more accurate understanding of the true dynamics of IAG.

As a minor contribution, our results can be extended straightforwardly to the smooth, non-convex setting, as demonstrated in Section 3.4. Overall, we anticipate that the simple modular nature of our proof framework can be built upon to develop tail bounds for more complex (e.g., accelerated, second-order) variance-reduced algorithms in the future.

**More Related Work.** The literature on stochastic VR algorithms is vast, and we refer the reader to the excellent survey by Gower et al. (2020). Aside from SAG and SAGA, other popular VR algorithms that enjoy linear convergence rates for smooth, strongly convex functions include SVRG (Johnson & Zhang, 2013), SDCA (Shalev-Shwartz & Zhang, 2013), S2GD (Konečný & Richtárik, 2017), MISO (Mairal, 2015), FINITO (Defazio et al., 2014b), and SARAH (Nguyen et al., 2017). Notably, different from the high-probability bounds we provide, these papers derive guarantees that hold in expectation. As such, our proof techniques differ from those in these works. For deterministic sampling, incremental gradient (IG) methods (Bertsekas et al., 2011) use only one component function to update the parameter in each iteration; like SGD, they suffer from slow sub-linear rates. Deterministic VR methods like IAG (Blatt et al., 2007) and DIAG (Mokhtari et al., 2018) use memory to achieve linear rates like GD. To our knowledge, no prior

work has unified the analysis of stochastic and deterministic VR methods within a single framework.

## 2. Technical Background

In this section, we introduce the two celebrated variance-reduction algorithms that we wish to study: SAG and SAGA. To that end, consider first-order iterative algorithms of the general form below for solving Problem (1):

$$x_{k+1} = x_k - \alpha g_k^{\mathcal{A}}, \qquad 0 \leq k \leq K - 1 \qquad (2)$$

where $\alpha \in (0, 1)$ is the step-size, $x_k \in \mathbb{R}^d$ denotes the parameter at iteration $k$, $g_k^{\mathcal{A}}$ is an estimate of the full gradient $\nabla f(x_k)$ at iteration $k$, $\mathcal{A}$ refers to the algorithm under study (e.g., GD, SGD, SAG, SAGA, etc.), and $K > 0$ denotes the horizon of the algorithm. Throughout the paper, we will assume that $x_0 = 0$ for clarity of exposition; our proofs naturally extend to the case where $x_0 \neq 0$ with routine modifications. We now discuss some relevant instances of (2), and their performance guarantees when each $f_i$ in (1) is $L$-smooth, and $f$ is $\mu$-strongly convex. Unless stated otherwise, this will be our running assumption on the functions that appear in (1).

• **GD**. In the GD algorithm, $g_k^{\text{GD}}$ takes the following form:

$$g_k^{\text{GD}} := \frac{1}{N} \sum_{i=1}^{N} \nabla f_i(x_k), \qquad (3)$$

which is essentially the **global (full) gradient** $\nabla f(x_k)$. By selecting $\alpha = 1/L$, the convergence rate for GD after $K$ iterations is (Bubeck et al., 2015):

$$f(x_K) - f(x^*) \leq \left(1 - \frac{1}{\kappa}\right)^K (f(x_0) - f(x^*)), \quad (4)$$

where $\kappa := L/\mu$ is the condition number. Although GD enjoys exact convergence to the optimum $x^*$ at a *linear* rate, it suffers from a significant computational bottleneck since $N$ gradients need to be computed at each iteration, where $N$ could be prohibitively large in practice.

• **SGD**. A well-studied alternative is SGD, which updates the parameter using a randomly selected component gradient, i.e., $g_k^{\text{SGD}} := \nabla f_{i_k}(x_k)$, where $i_k$ is sampled in an **I.I.D.** manner from $[N]$ *uniformly at random*. While SGD substantially reduces the per-iteration computational cost, the high variance of the update direction prevents convergence to the exact minimizer $x^*$ under a constant step-size. To ensure convergence to $x^*$, a diminishing step-size sequence is then needed, which leads to a slower *sub-linear* rate of $\mathcal{O}(1/K)$ (Bubeck et al., 2015; Moulines & Bach, 2011).

• **SAG and SAGA**. To reduce the variance of SGD, the SAG algorithm of Roux et al. (2012) maintains a memory of

previously computed component gradients. At each iteration $k$, SAG selects an index $i_k$ uniformly at random from $[N]$ as in SGD, computes the corresponding component gradient $\nabla f_{i_k}(x_k)$, and updates the iterate $x_k$ as per (2) using the average of all stored gradients, with

$$g_k^{\text{SAG}} := \frac{1}{N} \sum_{i=1}^{N} \nabla f_i(x_{\tau_{i,k}}) + \frac{\nabla f_{i_k}(x_k) - \nabla f_{i_k}(x_{\tau_{i_k,k}})}{N}.$$
$$(5)$$

Here, $\tau_{i,k} < k$ (initialized to $\tau_{i,0} = 0$ for all $i$) denotes the most recent iteration *before* iteration $k$ at which the component gradient $\nabla f_i$ was evaluated. If $\nabla f_i$ has not been accessed prior to iteration $k$, then $\nabla f_i(x_{\tau_{i,k}})$ is set to zero.

While SAG also computes only one component gradient per iteration as SGD, it manages to achieve linear convergence to $x^*$ using extra storage, with a rate given by:

$$\mathbb{E}[f(x_K)] - f(x^*) \leq \left(1 - \min\left\{\frac{1}{16\kappa}, \frac{1}{8N}\right\}\right)^K C_0,$$
$$(6)$$

where $C_0$ is a constant depending on initialization (Schmidt et al., 2017). While this was a remarkable result at the time, the convergence proof of SAG in Schmidt et al. (2017) is extremely complicated, and requires computer-aided tools to verify the non-negativity of certain polynomials that arise in their potential function. The difficulty in the analysis has often been attributed to the fact that for SAG, $g_k^{\text{SAG}}$ is not *unbiased*, and so the usual descent argument where the expected gradient error is zero does not apply.

To address this, the SAGA algorithm (Defazio et al., 2014a) preserves the sampling and memory mechanism of SAG, but proposes a bias-correction technique that yields an unbiased gradient estimator $g_k^{\text{SAGA}}$, defined as follows:

$$g_k^{\text{SAGA}} := \frac{1}{N} \sum_{i=1}^{N} \nabla f_i(x_{\tau_{i,k}}) + \nabla f_{i_k}(x_k) - \nabla f_{i_k}(x_{\tau_{i_k,k}}).$$
$$(7)$$

The parameter $x_k$ is then updated similarly following (2). The only difference of (7) from (5) is that the correction term $\nabla f_{i_k}(x_k) - \nabla f_{i_k}(x_{\tau_{i_k,k}})$ is added without the $1/N$ scaling factor, successfully removing the bias of the gradient estimator. This leads to a simpler convergence proof for SAGA relative to SAG. Moreover, the (in-expectation) convergence rate for SAGA in Defazio et al. (2014a) yields essentially the same bound as in (6) for SAG (up to constants).

**Paper Outline.** Although the update rules for SAG and SAGA in (5) and (7), respectively, are very similar, and they achieve essentially the same rates, their current analyses are dramatically different. In this context, perhaps surprisingly, we show that a unified proof can be developed to achieve high-probability bounds for both SAG and SAGA. This is the subject of Section 3. We also show that the simplicity of our proof lends itself to more general settings: we

consider non-convex objectives in Section 3.4 and Markov sampling in Section 3.5. Finally, in Section 4, we discuss how the proof technique developed in Section 3 for stochastic variance-reduced algorithms can be applied, *with no modifications at all*, to deterministic counterparts such as IAG. In the process, we significantly improve prior rates for IAG. Overall, our work sheds novel insights into the dynamics of celebrated variance-reduced algorithms that constitute the workhorse of modern machine learning problems.

# 3. Analysis and Results

In this section, we develop high-probability bounds for SAG and SAGA for smooth, strongly convex and non-convex objectives. Our proof has two key steps. In the first step (Section 3.1), we use Bernstein's inequality to bound a "gradient-staleness" effect that arises from sub-sampling. Informed by this bound, we construct a novel Lyapunov function and analyze its behavior in the second step (Section 3.3).

## 3.1. Step 1: Bounding the Staleness from Sub-Sampling

We start by noting that the main distinction between the GD gradient $g_k^{\mathbf{GD}}$ in (3) and the SAG/SAGA gradients $g_k^{\mathbf{SAG}}, g_k^{\mathbf{SAGA}}$ in (5) and (7) lies in the staleness of the component gradients in the latter, as captured by $\tau_{i,k}$. Our simple yet key observation is that while the staleness $\tau_{i,k}$ is random, it is not completely uncontrolled. In particular, using concentration, one can show that *the staleness of component gradients can be upper bounded with high probability*. This observation is formalized in the following lemma.

**Lemma 3.1** (Bounded Staleness). *For any $\delta \in (0, 1)$ and $\tau \geq (8N/3) \log(NK/\delta)$, with probability at least $1 - \delta$,*

$$k - \tau_{i,k} \leq \tau, \qquad \forall i \in [N],\ 0 \leq k \leq K - 1. \quad (8)$$

*Proof.* To prove Lemma 3.1, it suffices to first show that for any fixed component $i \in [N]$ and iteration $k = k_0$, the event that $i$ is not sampled within a window of $\tau$ consecutive iterations starting from $k_0$ occurs with probability at most $\delta/(NK)$. We then union-bound over all components and iterations. To this end, we will appeal to Bernstein's inequality, which we record below (Boucheron et al., 2003).

**Bernstein's Inequality.** *Let $X_1, \cdots, X_k$ be independent zero-mean random variables (RVs). Suppose that $|X_i| \leq M, \forall i \in [k]$. Then, for all $t > 0$, we have*

$$\mathbb{P}\left(\sum_{i=1}^{k} X_i \leq -t\right) \leq \exp\left(-\frac{\frac{1}{2}t^2}{\sum_{i=1}^{k} \mathbb{E}\left[X_i^2\right] + \frac{1}{3}Mt}\right). \quad (9)$$

With this tool at hand, let us fix a component $i \in [N]$, an iteration $k = k_0$, and define a random variable $Y_{i,k} := \mathbb{I}\{i_k = i\} \in \{0, 1\}$, where $\mathbb{I}$ is the indicator RV. Fix an integer $\tau > 0$ and note that *within any window of $\tau$ iterations*

starting from $k = k_0$, the probability that component $i$ is never sampled is given by $\mathbb{P}\left(\sum_{k=k_0}^{\tau+k_0-1} Y_{i,k} \leq 0\right)$.

The next thing to do is control this probability using Bernstein's inequality. Under the I.I.D. sampling model, we have $\mathbb{E}[Y_{i,k}] = p := 1/N$. Defining $X_{i,k} := Y_{i,k} - p$, let us create a sequence $\{X_{i,k}\}$ of zero-mean, bounded, and independent RVs with $\mathbb{E}[X_{i,k}] = 0$ and $|X_{i,k}| \leq M := 1$, for all $i \in [N]$ and $k \geq 0$. Furthermore, let us note that $\mathbb{E}[X_{i,k}^2] = \mathbb{V}[Y_{i,k}] = p(1 - p) \leq p$. Using (9), we have

$$
\begin{aligned}
\mathbb{P}\left(\sum_{k=k_0}^{\tau+k_0-1} Y_{i,k} \leq 0\right) &\overset{(a)}{=} \mathbb{P}\left(\sum_{k=k_0}^{\tau+k_0-1} X_{i,k} \leq -\tau p\right) \\
&\overset{(b)}{\leq} \exp\left(-\frac{\frac{1}{2}\tau^2 p^2}{\tau p + \frac{1}{3}\tau p}\right) \overset{(c)}{=} \exp\left(-\frac{3\tau}{8N}\right),
\end{aligned}
\quad (10)
$$

where in $(a)$ we use the definition of $X_{i,k}$, in $(b)$ we use (9) and the fact that $M = 1$, $\mathbb{E}[X_{i,k}^2] \leq p$, and in $(c)$, we use $p = 1/N$. Requiring the right-hand-side (RHS) of (10) to be smaller than a prescribed failure probability $\delta/(NK)$, and union-bounding over all $N$ components and $K$ iterations yields the desired claim of the lemma. $\square$

Informed by Lemma 3.1, we set $\tau = \lceil (8N/3) \log(NK/\delta)\rceil$, and define a "good event" $\mathcal{G}$ as follows:

$$\mathcal{G} := \{k - \tau_{i,k} \leq \tau,\ \forall i \in [N],\ \forall k \geq 0\}. \quad (11)$$

From Lemma 3.1, we know that $\mathcal{G}$ has measure at least $1-\delta$. Moreover, on event $\mathcal{G}$, the gradient estimators for SAG and SAGA use information at most $\tau$ time-steps old, allowing us to treat SAG and SAGA as methods with *uniformly bounded delay*. We will build on this insight for our subsequent analysis, where we will condition on the event $\mathcal{G}$.

## 3.2. Bounding the Gradient Error

First, let us define $e_k := g_k - \nabla f(x_k)$ as the error in the SAG/SAGA gradient estimator relative to the full gradient, where $g_k$ is either $g_k^{\mathbf{SAG}}$ or $g_k^{\mathbf{SAGA}}$ (we drop the superscript on $g_k^{\mathcal{A}}$ when it applies to both SAG and SAGA). Defining $r_k := f(x_k) - f(x^*)$ as the function sub-optimality gap, we have the following approximate descent lemma that captures the one-step descent in the function gap $r_k$; the result applies to both SAG and SAGA.

**Lemma 3.2** (Approximate Descent). *The following holds for both SAG and SAGA by selecting $\alpha \leq 1/(4L)$:*

$$r_{k+1} \leq r_k - \frac{\alpha}{4}\|\nabla f(x_k)\|_2^2 + \alpha\|e_k\|_2^2, \qquad \forall k \geq 0. \quad (12)$$

*Proof.* Using the smoothness of $f$, we have:

$$r_{k+1} \leq r_k + \langle \nabla f(x_k), x_{k+1} - x_k \rangle + \frac{L}{2} \|x_{k+1} - x_k\|_2^2$$

$$\overset{(a)}{\leq} r_k - \alpha \langle \nabla f(x_k), g_k \rangle + \frac{L\alpha^2}{2} \|g_k\|_2^2$$

$$\overset{(b)}{=} r_k - \alpha \langle \nabla f(x_k), \nabla f(x_k) + e_k \rangle + \frac{L\alpha^2}{2} \|\nabla f(x_k) + e_k\|_2^2$$

$$\leq r_k - \alpha \|\nabla f(x_k)\|_2^2 - \alpha \langle \nabla f(x_k), e_k \rangle$$
$$\quad + L\alpha^2 \left( \|\nabla f(x_k)\|_2^2 + \|e_k\|_2^2 \right)$$

$$\overset{(c)}{\leq} r_k - \left( \frac{\alpha}{2} - \alpha^2 L \right) \|\nabla f(x_k)\|_2^2 + \left( \frac{\alpha}{2} + \alpha^2 L \right) \|e_k\|_2^2.$$

Here, $(a)$ uses the update formula in (2), $(b)$ uses the definition of $e_k$, and $(c)$ uses Young's inequality for the inner product term. The claim then follows from $\alpha \leq 1/(4L)$. $\square$

Lemma 3.2 shows that the one-step descent is perturbed by the gradient error term $\|e_k\|_2^2$. The following lemma is then motivated by this issue, which provides an upper bound on $\|e_k\|_2^2$ that depends on *a window of past gradients*.

**Lemma 3.3** (Gradient Error). *On event $\mathcal{G}$, the gradient error $e_k$ satisfies the following for both SAG and SAGA:*

$$\|e_k\|_2^2 \leq 4L^2\tau\alpha^2 U_k, \ \ \forall k \geq \tau, \ \ \text{with } U_k := \sum_{j=1}^{\tau} \|g_{k-j}\|_2^2.$$

*Proof.* We first prove the result for SAG. Following the definition of $e_k$ and $g_k^{\text{SAG}}$ in (5), we have

$$\|e_k\|_2 = \frac{1}{N} \left\| \sum_{i \neq i_k} \nabla f_i(x_{\tau_{i,k}}) + \nabla f_{i_k}(x_k) - \sum_{i \in [N]} \nabla f_i(x_k) \right\|_2$$

$$= \frac{1}{N} \left\| \sum_{i \neq i_k} \left( \nabla f_i(x_{\tau_{i,k}}) - \nabla f_i(x_k) \right) \right\|_2$$

$$\overset{(a)}{\leq} \frac{1}{N} \sum_{i \neq i_k} \|\nabla f_i(x_{\tau_{i,k}}) - \nabla f_i(x_k)\|_2$$

$$\overset{(b)}{\leq} \frac{1}{N} \sum_{i \neq i_k} L \|x_{\tau_{i,k}} - x_k\|_2$$

$$\overset{(c)}{\leq} \frac{L}{N} \sum_{i \neq i_k} \sum_{j=1}^{\tau} \|x_{k-j+1} - x_{k-j}\|_2$$

$$\overset{(d)}{\leq} L\alpha \sum_{j=1}^{\tau} \|g_{k-j}^{\text{SAG}}\|_2,$$

(13)

where $(a)$ is due to the triangle inequality, $(b)$ uses the smoothness of $f_i$, $(c)$ uses the triangle inequality and the fact that delays are at most $\tau$ conditioned on $\mathcal{G}$ (from Lemma 3.1), and $(d)$ follows from (2). Squaring both sides

of (13) and using Jensen's inequality yields the desired claim for SAG.

Similarly, for SAGA, we have

$$\|e_k\|_2 \leq \left\| \frac{1}{N} \sum_{i=1}^{N} \left( \nabla f_i(x_{\tau_{i,k}}) - \nabla f_i(x_k) \right) \right\|_2$$
$$\quad + \left\| \nabla f_{i_k}(x_k) - \nabla f_{i_k}(x_{\tau_{i_k,k}}) \right\|_2$$

$$\leq 2L\alpha \sum_{j=1}^{\tau} \|g_{k-j}^{\text{SAGA}}\|_2,$$

(14)

where we omit the intermediate steps since they follow exactly as in the SAG analysis in (13). $\square$

Note that Lemma 3.3 holds only for $k \geq \tau$ such that the past gradient terms in $U_k$ are well defined. The next corollary is an immediate consequence of Lemma 3.3 that follows from the definition of $e_k$ and Jensen's inequality.

**Corollary 3.4** (Gradient Bound). *On event $\mathcal{G}$, the following holds for both SAG and SAGA, for all $k \geq \tau$:*

$$\|g_k\|_2^2 \leq 2 \|\nabla f(x_k)\|_2^2 + 8L^2\tau\alpha^2 U_k.$$

(15)

### 3.3. Step 2: Designing the Lyapunov Function

From Lemma 3.3, observe that the bound on $\|e_k\|_2^2$ depends on a window of past gradients, and hence, cannot be absorbed by a single-step descent argument as in (12). This motivates the choice of a Lyapunov function with a *shifted window* term that tracks the recent history. To that end, we construct the following Lyapunov function $V_k$ for $k \geq \tau$:

$$V_k := f(x_k) - f(x^*) + L\alpha^2 W_k,$$
$$\text{with } W_k := \sum_{j=1}^{\tau} (\tau - j + 1) \|g_{k-j}\|_2^2.$$

(16)

The weight $\tau - j + 1$ assigned to the past gradient $\|g_{k-j}\|_2^2$ is motivated by how past gradients accumulate in the one-step descent analysis. Specifically, for a fixed $k$, from the descent inequality in Lemma 3.2, and the bound on $\|e_k\|_2^2$ from Lemma 3.3, we note that the stale gradient $\|g_{k-j}\|_2^2$ appears in exactly $\tau - j + 1$ future descent inequalities over the window $[k, k + \tau - 1]$. By assigning larger weights to more recent gradients, our choice of the Lyapunov function in (16) carefully accounts for this multiplicity. In a moment, we will see how this choice allows us to establish a one-step contractive recursion for $V_k$. To proceed, we will need the following facts about the "shifted window" terms $U_k$ and $W_k$ associated with the Lyapunov function.

**Fact 3.5.** *The following holds for any $k \geq \tau$:*

$$W_{k+1} = W_k - U_k + \tau \|g_k\|_2^2.$$

(17)

**Fact 3.6.** *The following holds for any $k \geq \tau$:*

$$W_k \leq \tau U_k. \tag{18}$$

The proofs of these facts follow directly from the definitions of $U_k$ and $W_k$, and are hence omitted. We now have all the pieces needed to establish a one-step recursion for $V_k$.

**Lemma 3.7** (One-Step Recursion). *Suppose $f$ in (1) is $\mu$-strongly convex. Let $\alpha = 1/(16L\tau)$. Then, on event $\mathcal{G}$, the following is true for both SAG and SAGA:*

$$V_{k+1} \leq \left(1 - \frac{\alpha\mu}{4}\right) V_k, \quad \forall k \geq \tau. \tag{19}$$

*Proof.* Using (16), plugging the bound on $\|e_k\|_2^2$ in Lemma 3.3 into (12), and adding $L\alpha^2 W_{k+1}$ to both sides of (12) yields the following recursion that holds for both SAG and SAGA:

$$
\begin{aligned}
V_{k+1} &\leq r_k - \frac{\alpha}{4} \|\nabla f(x_k)\|_2^2 + 4L^2\tau\alpha^3 U_k + L\alpha^2 W_{k+1} \\
&\overset{(a)}{=} r_k - \frac{\alpha}{4} \|\nabla f(x_k)\|_2^2 + 4L^2\tau\alpha^3 U_k \\
&\quad + L\alpha^2 \left(W_k - U_k + \tau \|g_k\|_2^2\right) \\
&\overset{(b)}{\leq} r_k - \frac{\alpha}{4} \|\nabla f(x_k)\|_2^2 + 4L^2\tau\alpha^3 U_k \\
&\quad + L\alpha^2 \left(W_k - U_k + 2\tau \|\nabla f(x_k)\|_2^2 + 8L^2\tau^2\alpha^2 U_k\right) \\
&= r_k - \left(\frac{\alpha}{4} - 2L\tau\alpha^2\right) \|\nabla f(x_k)\|_2^2 + L\alpha^2 W_k \\
&\quad + \left(4L^2\tau\alpha^3 + 8L^3\tau^2\alpha^4 - L\alpha^2\right) U_k \\
&\overset{(c)}{\leq} r_k - \frac{\alpha}{8} \|\nabla f(x_k)\|_2^2 + L\alpha^2 \left(W_k - \frac{1}{2} U_k\right) \\
&\overset{(d)}{\leq} \left(1 - \frac{\alpha\mu}{4}\right) r_k + L\alpha^2 \left(1 - \frac{1}{2\tau}\right) W_k \\
&\overset{(e)}{\leq} \left(1 - \frac{\alpha\mu}{4}\right) r_k + L\alpha^2 \left(1 - \frac{\alpha\mu}{4}\right) W_k \\
&= \left(1 - \frac{\alpha\mu}{4}\right) V_k.
\end{aligned}
\tag{20}
$$

Here, $(a)$ follows from decomposing $W_{k+1}$ using Fact 3.5, $(b)$ uses Corollary 3.4 to bound $\|g_k\|_2^2$, $(c)$ holds by selecting $\alpha = 1/(16L\tau)$ such that $2L\tau\alpha^2 \leq \alpha/8$, $4L^2\tau\alpha^3 \leq L\alpha^2/4$, and $8L^3\tau^2\alpha^4 \leq L\alpha^2/4$, $(d)$ uses Fact 3.6 to bound $U_k$ and the gradient domination property of strong convexity, and $(e)$ holds by selecting $\alpha = 1/(16L\tau) \leq 2/(\mu\tau)$. $\square$

There is an important caveat here after obtaining inequality (19). Since the bound on $\|e_k\|_2^2$ in Lemma 3.3 only holds for $k \geq \tau$, the one-step Lyapunov recursion in Lemma 3.7 therefore only makes sense for $k \geq \tau$. As such, iterating (19) from $k = \tau$ to $k = K - 1$ yields:

$$f(x_K) - f(x^*) \overset{(*)}{\leq} V_K \leq \left(1 - \frac{\alpha\mu}{4}\right)^{K-\tau} V_\tau, \tag{21}$$

where $(*)$ holds from the definition of $V_K$. The appearance of $V_\tau$ reflects a finite burn-in period where the bounded staleness condition has not yet kicked in. To complete the analysis, we need to argue that at the end of this period, $V_\tau$ remains bounded. This is the subject of the next result.

**Lemma 3.8** (Burn-In Effects). *Define $B := \max\{\|x^*\|_2, \|x_1^*\|_2, \cdots \|x_N^*\|_2\}$, where $x_i^* \in \arg\min_{x \in \mathbb{R}^d} f_i(x)$. With $\alpha = 1/(16L\tau)$, the following is true: $V_\tau \leq 3LB^2$.*

The dependence of $B$ on the minimizers $\{x_i^*\}$ of the component functions can be intuitively explained as follows. Recall that $\tau = \tilde{\mathcal{O}}(N)$, i.e., the initial burn-in period is on the order of the number of components $N$ (up to log factors). As such, there are bound to be certain time-steps $k < \tau$, such that, at time $k$, not every component function has been sampled at least once. Nonetheless, for both SAG and SAGA, updates to the iterates are still made during the initial burn-in period. As a result, the effective function being optimized during this phase can differ from the true one $f$, and the iterates may get biased toward the minimizers of the component functions. A possible remedy is to modify how the algorithm operates during the burn-in phase so that no iterate updates are performed during the first $\tau$ iterations, and the algorithm only collects component gradients and updates the memory during this phase. Iterate updates begin after $\tau$, once all components have been sampled at least once on the event $\mathcal{G}$. In this case, one can show that $B = \|x^*\|_2$ suffices to capture burn-in effects. The details are provided in Appendix B.2. When the initial condition $x_0 \neq 0$, one can derive similar bounds as in Lemma 3.8 by modifying the definition of $B$ to include $\|x_0\|_2$. The proof of this follows identical steps to that of Lemma 3.8 in Appendix B.1.

The proof of Lemma 3.8 can be divided into three steps: (i) we show that during iterations $0 \leq k \leq \tau$, the iterates are bounded by $\|x_k\|_2 \leq B$ using induction; (ii) using this, we show that the gradient norms for $0 \leq k \leq \tau$ are bounded by $\|g_k\|_2 \leq 6LB$; and (iii) finally, we show that $V_\tau$ is bounded by $3LB^2$ using (i), (ii), and the definition of $V_\tau$. The details of the proof are provided in Appendix B.1.

We now resume our analysis. Plugging the bound for $V_\tau$ in Lemma 3.8 into (21), and selecting $\alpha = 1/(16L\tau)$ yields

$$
\begin{aligned}
f(x_K) - f(x^*) &\leq 3LB^2 \left(1 - \frac{1}{64\tau\kappa}\right)^K \left(1 - \frac{1}{64\tau\kappa}\right)^{-\tau} \\
&\leq 6LB^2 \left(1 - \frac{1}{64\tau\kappa}\right)^K.
\end{aligned}
$$

In the last step, we used Bernoulli's inequality: $(1 + x)^r \geq 1 + rx$, where $r \geq 1$ is a positive integer and $x \geq -1$. Based on our developments in Sections 3.1– 3.3, and the fact that event $\mathcal{G}$ has measure at least $1 - \delta$, we have established the following result.

**Theorem 3.9** (SAG/SAGA, Strongly Convex Case)**.** *Suppose that each $f_i$ in (1) is $L$-smooth, and $f$ is $\mu$-strongly convex. Given any $\delta \in (0,1)$, let $\tau = \lceil (8N/3) \log(NK/\delta) \rceil$, and set $\alpha = 1/(16L\tau)$. Then, with probability at least $1-\delta$, the following holds for both SAG and SAGA when $K > \tau$:*

$$f(x_K) - f(x^*) \leq 6LB^2 \left( 1 - \frac{1}{64\tau\kappa} \right)^K, \quad (22)$$

*where $\kappa = L/\mu$ and $B$ is as defined in Lemma 3.8.*

**Main Takeaways.** Our result above reveals that with high-probability, SAG and SAGA converge exponentially fast to the optimal point $x^*$, where the exponent depends on the product of the condition number $\kappa$ and the staleness factor $\tau$. As far as we are aware, Theorem 3.9 *is the first high-probability bound that applies identically to both SAG and SAGA*. Notably, our analysis that leads to this result is significantly simpler, shorter, and self-contained compared to the highly involved and computer-aided analysis for SAG in Schmidt et al. (2017). Since $\tau = \tilde{\mathcal{O}}(N)$, the exponent of convergence in (22) is slower by a factor of $N$ relative to the exponent for gradient descent in (4). Intuitively, this makes sense since $N$ iterations of SAG/SAGA lead to the same number of gradient evaluations as in one step of GD.

Although the rate in (6) is better than that in (22), we conjecture that this difference arises from the fact that the former is an in-expectation guarantee, while the latter is a high-probability bound. In particular, high-probability bounds need to account for tail events where the delay can indeed be on the order $\tilde{\mathcal{O}}(N)$. To provide further insights about our rate, consider the deterministic delayed GD update rule of the form $x_{k+1} = x_k - \alpha \nabla f(x_{k-\tau})$, where $\tau > 0$ is a constant delay. Interestingly, for this update rule, it is shown by Arjevani et al. (2020) that a rate on the order of $\exp(-K/(\tau\kappa))$ is, in fact, *tight*. In other words, the deterioration of the exponent by the delay $\tau$ (relative to GD) is unavoidable. On a related note, observe that the logarithmic dependence on the failure probability $\delta$ appears in the exponent in (22) as opposed to a pre-factor in typical high-probability bounds. This can be attributed to the fact that the delay $\tau$ depends on $\log(1/\delta)$, and the appearance of the delay $\tau$ in the exponent seems unavoidable.

### 3.4. Extension to Non-Convex Objectives

We now show that the analysis for the strongly convex case can be easily generalized to account for non-convex objectives. To see this, observe that just on the basis of smoothness of each $f_i$, one can arrive at inequality $(c)$ of the chain of inequalities in (20). We then have

$$\begin{aligned} V_{k+1} &\leq r_k - \frac{\alpha}{8} \|\nabla f(x_k)\|_2^2 + L\alpha^2 \left( W_k - \frac{1}{2} U_k \right) \\ &\leq V_k - \frac{\alpha}{8} \|\nabla f(x_k)\|_2^2, \end{aligned} \quad (23)$$

where the second inequality follows from discarding the negative term $-L\alpha^2 U_k/2$. Rearranging and telescoping (23) from iteration $k = \tau$ to $k = K - 1$ yields

$$\frac{1}{K-\tau} \sum_{k=\tau}^{K-1} \|\nabla f(x_k)\|_2^2 \leq \frac{8V_\tau}{(K-\tau)\alpha} \leq \frac{256L\tau V_\tau}{K}, \quad (24)$$

when $K \geq 2\tau$, and $\alpha = 1/(16L\tau)$. Using the bound on $V_\tau$ from Lemma 3.8 immediately yields the following result.

**Theorem 3.10** (SAG/SAGA, Non-Convex Case)**.** *Suppose that each $f_i$ in (1) is $L$-smooth. Given any $\delta \in (0,1)$, let $\tau$ and $\alpha$ be as in Theorem 3.9. Then, with probability at least $1 - \delta$, the following holds for both SAG and SAGA:*

$$\frac{1}{K-\tau} \sum_{k=\tau}^{K-1} \|\nabla f(x_k)\|_2^2 \leq \frac{768L^2B^2\tau}{K}, \forall K \geq 2\tau. \quad (25)$$

**Main Takeaway**. For smooth, non-convex objectives, it is well known that gradient descent provides a $\mathcal{O}(1/K)$ convergence rate for the object on the LHS of (25) (Bubeck et al., 2015). Interpreting $K/\tau$ as the "effective" number of iterations (due to sub-sampling), Theorem 3.10 establishes a similar high-probability rate for SAG and SAGA, and complements in-expectation guarantees for these algorithms under non-convex objectives, established in Reddi et al. (2016b;a); Koloskova et al. (2023).

### 3.5. Extension to Markov Sampling

Thus far, we have worked under the assumption that at each iteration $k$, a component $i_k$ is sampled in an I.I.D. manner, uniformly at random from $[N]$. In this section, we will significantly relax such an assumption, and demonstrate that our analysis seamlessly extends to a more general Markov sampling scheme. Specifically, we now consider a scenario where the sampling indices $\{i_k\}_{k \geq 0}$ form the trajectory of a time-homogeneous, aperiodic, and irreducible Markov chain $\mathcal{M}$ supported on $[N]$. Let $\pi$ be the stationary distribution of this ergodic chain, and, for simplicity, assume that $i_0 \sim \pi$, causing the chain to be stationary.[1]

We note that the basic SGD algorithm has been analyzed under Markov sampling in several papers; for instance, see Duchi et al. (2012); Sun et al. (2018); Doan (2022). Like us, these papers also work under the assumption that the data-generating Markov chain is ergodic. However, to our knowledge, there are no known high-probability bounds for variance-reduced algorithms such as SAG and SAGA under Markov sampling. Our goal is to close this gap.

With this in mind, let $\pi_{\min} := \min_{i \in [N]} \pi_i > 0$ denote the smallest entry in the stationary distribution $\pi$, representing the *minimum visitation probability*. It should be noted

---

[1]The assumption of stationarity can be avoided at the expense of more algebra that we omit here for clarity of exposition.

that for our subsequent analysis, we do not require $\pi$ to be a uniform distribution over $[N]$. As such, our analysis can handle the case when the gradient estimators (for both SAG and SAGA) are *biased*. To build some intuition, let us think back to the analysis under I.I.D. sampling. More than the I.I.D. aspect itself, what mattered was the fact that, with high probability, each component function is visited sufficiently often. This, in turn, ensured a bounded staleness effect, which caused the rest of the analysis to go through. Thus, as long as we can argue that under Markov sampling, a similar bounded staleness property is preserved, the remainder of the analysis will be identical to the I.I.D. case. We now show that ergodicity buys us exactly this desired property. To that end, we introduce the *mixing time* function of $\mathcal{M}$ following Dorfman & Levy (2022): $d_{mix}(k) := \sup_{i \in [N]} D_{TV}\left(\mathbb{P}(i_k \in \cdot \mid i_0 = i), \pi\right)$, where $D_{TV}$ is the *total variation distance* between probability measures. Next, we define the mixing time of $\mathcal{M}$ as $t_{mix} := \inf\{k \mid d_{mix}(k) \leq 1/4\}$. Using the objects defined above, we can then prove the following key result.

**Lemma 3.11** (Bounded Staleness under Markov Sampling). *For any $\delta \in (0,1)$ and $\tau \geq (88t_{mix}/\pi_{\min})\log(NK/\delta)$, the following holds with probability at least $1 - \delta$,*

$$k - \tau_{i,k} \leq \tau, \qquad \forall i \in [N], \ 0 \leq k \leq K - 1. \quad (26)$$

The proof of Lemma 3.11 is provided in Appendix C; the key technical tool we use to establish this result is a variant of Bernstein's inequality for Markov sampling developed by Paulin (2015). The only distinction between Lemma 3.11 and Lemma 3.1 lies in the dependence of $\tau$ on the minimum visitation probability $\pi_{\min}$ and the mixing time $t_{mix}$. Informed by Lemma 3.11, define the new staleness parameter as $\tau = \lceil (88t_{mix}/\pi_{\min})\log(NK/\delta) \rceil$. Now suppose the good event $\mathcal{G}$ in (11) is defined exactly as before with this new choice of $\tau$. Conditioned on this event $\mathcal{G}$, the remainder of the analysis under Markov sampling is identical to that under I.I.D. sampling carried out in Sections 3.2 and 3.3. As a result, we immediately obtain the following theorem.

**Theorem 3.12** (SAG/SAGA, Markov Sampling). *Consider the Markov sampling scheme described in Section 3.5. Suppose that each $f_i$ in (1) is $L$-smooth, and $f$ is $\mu$-strongly convex. Given any $\delta \in (0,1)$, let $\tau = \lceil (88t_{mix}/\pi_{\min})\log(NK/\delta) \rceil$, and set $\alpha = 1/(16L\tau)$. Then, with probability at least $1 - \delta$, the following holds for both SAG and SAGA when $K > \tau$:*

$$f(x_K) - f(x^*) \leq 6LB^2 \left(1 - \frac{1}{64\tau\kappa}\right)^K. \quad (27)$$

**Main Takeaways.** The main takeaway is that our result above under Markov sampling matches that for the I.I.D. case, except for the fact that the staleness parameter $\tau$ now depends on the mixing time and the minimum visitation

probability of the Markov chain. Essentially, up to log factors, one can now interpret $K/t_{\text{cov}}$ as the effective number of iterations, where $t_{\text{cov}} := t_{mix}/\pi_{min}$. Since Theorem 3.12 appears to be the *first high-probability bound for SAG and SAGA under Markov sampling*, we cannot comment on the tightness of our bound in terms of its dependence on $t_{cov}$. That said, the inflation by a factor of $t_{\text{cov}}$ is typically seen for stochastic approximation algorithms subject to Markov sampling, when the Markov chain is supported on a finite state-space; for instance, for tabular Q-learning, see Qu & Wierman (2020). The inflation by the mixing time $t_{mix}$ appears more generally for SGD in Nagaraj et al. (2020), for RL algorithms like temporal-difference learning in Bhandari et al. (2018); Srikant & Ying (2019); Mitra (2024), and nonlinear stochastic approximation in Chen et al. (2022).

**Remark 3.13.** *All our bounds thus far require prior knowledge of the horizon $K$ to inform the delay parameter $\tau$, which, in turn, dictates the choice of the step-size $\alpha$. We conjecture that it should be possible to derive bounds without prior knowledge of $K$ by leveraging the "doubling-trick" from the bandit's literature (Lattimore & Szepesvári, 2020).*

# 4. Extension to the IAG Method

Although we have considered stochastic sampling schemes thus far, we now show that our analysis framework can easily accommodate deterministic sampling patterns, as well. To that end, we consider the classical **incremental aggregated gradient (IAG)** method, a *deterministic* counterpart of SAG, introduced by Blatt et al. (2007). The gradient estimator $g_k^{\textbf{IAG}}$ of IAG has the same aggregated form as SAG in (5), and the iterate is also updated via (2). The key difference is that the component functions are sampled one at a time in *any* deterministic order, such that every component is sampled at least once in every $\tau$ iterations, i.e., we have

$$k > \tau_{i,k} \geq k - \tau, \quad (28)$$

where $\tau \in \mathbb{N}^+$ is some prescribed parameter, and $\tau_{i,k}$ has the same meaning as before. This condition coincides exactly with the high probability event $\mathcal{G}$ in (11), where the maximum delay in sampling any component is $\tau$. As a result, for the IAG algorithm, event $\mathcal{G}$ occurs with probability 1. Consequently, Theorems 3.9 and 3.10 hold for IAG **deterministically** without any modification. We record this observation below for the strongly convex case.

**Theorem 4.1** (IAG, Strongly Convex Case). *Suppose that each $f_i$ in (1) is $L$-smooth, and $f$ is $\mu$-strongly convex. Consider the IAG method with a sampling pattern that satisfies (28). With $\alpha = 1/(16L\tau)$, the following holds:*

$$f(x_K) - f(x^*) \leq 6LB^2 \left(1 - \frac{1}{64\tau\kappa}\right)^K, \forall K > \tau. \quad (29)$$

**Main Takeaways.** Comparing Theorem 4.1 with Theo-

rem 3.9, we note that the deterministic guarantee for IAG is *identical* to the high-probability bound we derived earlier for SAG/SAGA. Such a finding appears to be new. Perhaps more interestingly, our developments so far reveal that a single proof technique suffices to provide a unified treatment of both stochastic and deterministic variance-reduced algorithms. In addition to this unification, a key contribution of our work is that it significantly improves upon the best known convergence rate for IAG, as we discuss below.

*Tighter bounds for IAG.* As far as we are aware, the best known rate for the IAG method for smooth and strongly convex objectives was derived by Gurbuzbalaban et al. (2017), and is as follows:

$$f(x_K) - f(x^*) \leq \frac{L}{2} \left( 1 - \frac{c_\tau}{(\kappa + 1)^2} \right)^{2K} \|x_0 - x^*\|_2^2,$$

where $c_\tau := 2/(25\tau(2\tau + 1))$. From the above display, we note that while the convergence is still exponential, the exponent scales *quadratically* in both the condition number $\kappa$, and the delay $\tau$. In sharp contrast, our analysis for IAG in Theorem 4.1 is able to achieve a *linear* dependence in both $\kappa$ and $\tau$. This is a significant improvement for ill-conditioned problems. Moreover, note that for a simple cyclic sampling pattern, $\tau = N - 1$. Thus, for modern ERM problems, where $N$ represents a potentially large number of data samples, our rate marks a considerable improvement over prior work. Notably, such an improvement is a free byproduct of our unified proof strategy, and requires no extra work beyond what we did for SAG/SAGA.

## 5. Conclusion and Future Work

We introduced a novel *unified* proof framework that yields linear convergence rates for celebrated variance-reduced algorithms such as SAG, SAGA, and IAG. In the process, we provided the first high-probability bounds for SAG and SAGA, and significantly sharpened known rates for IAG. Finally, we showed that our proof techniques can easily accommodate Markov sampling schemes.

While the point of this paper was to argue that a single proof technique applies identically to both SAG and SAGA, our analysis framework is by no means limited to just these two algorithms. Indeed, for *single-loop* VR algorithms with stochastic sub-sampling, we anticipate that much of our ideas will carry through. For instance, the "bounded delay" event (Lemma 3.1), where the staleness of all component gradients is bounded, is a property of the sampling scheme and not the particular algorithm. As a result, Lemma 3.1 would be directly applicable to *any* single-loop VR algorithm under I.I.D. sub-sampling. Conditioned on the "bounded-delay event", we conjecture that the rest of the analysis should follow similarly for other VR methods, where we first establish a one-step descent (Lemma 3.2),

and bound the gradient error (Lemma 3.3). The latter would then inform the choice of the Lyapunov function for the specific VR method under study. As future work, we plan to verify this conjecture by applying our proof framework to broader classes of single-loop methods (e.g., second-order, proximal, accelerated, distributed, etc.)

On a related note, *double-loop* methods such as SVRG (Johnson & Zhang, 2013) and SARAH (Nguyen et al., 2017) periodically reset their reference points and gradient estimators. In this case, the staleness is *deterministically* bounded by the outer-loop length, eliminating the need for a high-probability bounded-delay event. Although a Lyapunov analysis similar to that pursued in this paper may still be possible, it is unclear whether new insights would emerge from it.

## Acknowledgments

This work is partially funded by the following grants from the National Science Foundation: NSF-CCF-2225555 and NSF CAREER award 2542396.

## Impact Statement

This paper presents work whose goal is to advance the field of Machine Learning. We cannot think of any potential societal consequences of our work that need to be specifically highlighted here.

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

# A. Useful Facts

To keep the paper self-contained, we recall the basic definitions and implications of smoothness and strong-convexity used in the main text. For proofs of the implications, we refer the reader to Bubeck et al. (2015).

**Definition A.1** (Smoothness). A function $f : \mathbb{R}^d \to \mathbb{R}$ is $L$-smooth if for any $x, y \in \mathbb{R}^d$, the following holds:

$$\|\nabla f(x) - \nabla f(y)\|_2 \le L \|x - y\|_2 , \tag{30}$$

where $\nabla(\cdot)$ denotes the gradient operator.

An immediate consequence of smoothness is the following:

$$f(y) - f(x) \le \langle y - x, \nabla f(x) \rangle + \frac{L}{2} \|y - x\|^2, \forall x, y \in \mathbb{R}^d. \tag{31}$$

**Definition A.2** (Strong Convexity). A function $f : \mathbb{R}^d \to \mathbb{R}$ is $\mu$-strongly convex if the following holds for any $x, y \in \mathbb{R}^d$:

$$f(y) \ge f(x) + \langle \nabla f(x), y - x \rangle + \frac{\mu}{2} \|x - y\|_2^2 . \tag{32}$$

The following gradient-domination property is a consequence of strong-convexity:

$$\|\nabla f(y)\|^2 \ge 2\mu(f(y) - f(x^*)), \forall y \in \mathbb{R}^d. \tag{33}$$

# B. Proof of Lemma 3.8

In this section, we provide proofs of Lemma 3.8 under two cases: (i) the original scheme for SAG/SAGA where updates are made during the initial burn-in period, and (ii) a modified scheme where no iterate updates are performed during the first $\tau$ iterations. As we shall see, the original scheme incurs dependence on the local minimizers of the component functions due to updates towards biased directions, while the modified scheme only depends on the global minimizer of $f$ as is standard with in-expectation analyses.

## B.1. Original Scheme

As stated immediately after Lemma 3.8, the proof is divided into three steps: (i) proving bounded iterates, (ii) proving bounded gradients, and (iii) proving bounded $V_\tau$. To this end, we first state the following claim implying that the iterates are deterministically bounded for $0 \le k \le \tau$:

**Claim B.1** (Bounded Iterates). *Fix $0 \le k \le \tau$, we claim that the following holds for both SAG and SAGA:*

$$\|x_k\|_2 \le B, \tag{34}$$

*where $B$ is as defined in Lemma 3.8.*

*Proof.* We prove this claim by induction and first focus on the analysis of SAG. The base case holds trivially by initializing $x_0 = 0$. Suppose that this claim holds up to iteration $0 \le k \le \tau - 1$. Then for iteration $k + 1$, we have

$$
\begin{aligned}
x_{k+1} &\overset{(a)}{=} x_k - \frac{\alpha}{N} \left( \sum_{i \ne i_k} \nabla f_i(x_{\tau_{i,k}}) + \nabla f_{i_k}(x_k) \right) \\
&\overset{(b)}{=} x_k - \alpha \nabla f(x_k) - \frac{\alpha}{N} \left( \sum_{i \ne i_k} \nabla f_i(x_{\tau_{i,k}}) + \nabla f_{i_k}(x_k) - \sum_{i=1}^N \nabla f_i(x_k) \right) \\
&= x_k - \alpha \nabla f(x_k) - \frac{\alpha}{N} \sum_{i \ne i_k} \left( \nabla f_i(x_{\tau_{i,k}}) - \nabla f_i(x_k) \right),
\end{aligned}
\tag{35}
$$

where $(a)$ uses the definition of $g_k^{\mathbf{SAG}}$, and $(b)$ holds by adding and subtracting $\nabla f(x_k)$.

Taking the 2-norm on both sides and using triangle inequality yields

$$
\begin{aligned}
\|x_{k+1}\|_2 &\le \|x_k\|_2 + \alpha \|\nabla f(x_k) - \nabla f(x^*)\|_2 + \frac{\alpha}{N} \sum_{i \ne i_k} \|\nabla f_i(x_{\tau_{i,k}}) - \nabla f_i(x_k)\|_2 \\
&\overset{(a)}{\le} \|x_k\|_2 + \alpha L \|x_k - x^*\|_2 + \frac{\alpha}{N} \sum_{i \in \mathcal{C}_k, i \ne i_k} L \|x_{\tau_{i,k}} - x_k\|_2 + \frac{\alpha}{N} \sum_{i \in [N] \backslash \mathcal{C}_k, i \ne i_k} \|\nabla f_i(x_k) - \nabla f_i(x_i^*)\|_2 \\
&\overset{(b)}{\le} \|x_k\|_2 + \alpha L (\|x_k\|_2 + \|x^*\|_2) + \frac{\alpha}{N} \sum_{i \in \mathcal{C}_k, i \ne i_k} L (\|x_{\tau_{i,k}}\|_2 + \|x_k\|_2) + \frac{\alpha}{N} \sum_{i \in [N] \backslash \mathcal{C}_k, i \ne i_k} L (\|x_k\|_2 + \|x_i^*\|_2) \\
&\overset{(c)}{\le} (1 + L\alpha) \|x_k\|_2 + 3LB\alpha,
\end{aligned}
\tag{36}
$$

where $(a)$ uses smoothness, and $\mathcal{C}_k$ denotes the set of component indices that have been sampled at least once up to iteration $k$. The reason for this definition is that in the first inequality, $\nabla f_i(x_{\tau_{i,k}})$ might be set to 0 since it is possible that component $i$ has never been accessed up to iteration $k$. Inequality $(b)$ holds due to smoothness and the fact that $\nabla f_i(x_i^*) = 0$. Inequality $(c)$ uses the definition of $B$ and the induction hypothesis up to iteration $k$.

Iterating (36) for $k + 1$ steps from $k = 0$ yields

$$
\|x_{k+1}\|_2 \le (1 + L\alpha)^{k+1} \|x_0\|_2 + 3LB\alpha \sum_{j=0}^k (1 + L\alpha)^{k-j}
\tag{37}
$$

$$
\le 3LB\alpha \cdot \tau (1 + L\alpha)^\tau \le 4LB\tau\alpha \le \frac{B}{4} \le B,
$$

where we use the fact that $x_0 = 0$, $k + 1 \leq \tau$, $1 + x \leq e^x$, and $\alpha \leq 1/(16L\tau)$. The proof is then complete for the SAG case.

The proof for the case of SAGA carries through similarly as in the proof of Lemma 3.3. Specifically, the one-step descent of SAGA can be written as

$$x_{k+1} = x_k - \alpha \nabla f(x_k) - \frac{\alpha}{N} \sum_{i=1}^{N} \left( \nabla f_i(x_{\tau_{i,k}}) - \nabla f_i(x_k) \right) - \alpha \left( \nabla f_{i_k}(x_k) - \nabla f_{i_k}(x_{\tau_{i_k,k}}) \right). \tag{38}$$

Taking the 2-norm on both sides and using triangle inequality yields

$$\|x_{k+1}\|_2 \leq \|x_k\|_2 + \frac{\alpha}{N} \sum_{i=1}^{N} \left\| \nabla f_i(x_{\tau_{i,k}}) - \nabla f_i(x_k) \right\|_2 + \alpha \left\| \nabla f_{i_k}(x_k) - \nabla f_{i_k}(x_{\tau_{i_k,k}}) \right\|_2 + \alpha \left\| \nabla f(x_k) - \nabla f(x^*) \right\|_2$$
$$\leq (1 + L\alpha) \|x_k\|_2 + 5LB\alpha. \tag{39}$$

The rationale is similar to the SAG case and is hence omitted, where one needs to break the analysis into two cases: (i) component $i$ has been sampled before iteration $k$, and (ii) it has never been sampled before iteration $k$. As has been shown for the SAG analysis, both cases yield the exact same bound. Iterating (39) for $k + 1$ steps from $k = 0$ yields the same bound. The proof is then complete.

$\square$

The next claim provides an upper bound on the gradients $\|g_k\|_2$ when $0 \leq k \leq \tau$.

**Claim B.2** (Bounded Gradients). *Fix $0 \leq k \leq \tau$, the following holds for both SAG and SAGA:*

$$\|g_k\|_2 \leq 6LB. \tag{40}$$

*Proof.* For SAG, we can write

$$\left\| g_k^{\mathbf{SAG}} \right\|_2 \leq \frac{1}{N} \sum_{i \neq i_k} \left\| \nabla f_i(x_{\tau_{i,k}}) - \nabla f_i(x_k) \right\|_2 + \left\| \nabla f(x_k) - \nabla f(x^*) \right\|_2$$
$$\leq 2LB + 2LB = 4LB \leq 6LB, \tag{41}$$

where we used smoothness and Claim B.1.

For SAGA, we can write

$$\left\| g_k^{\mathbf{SAGA}} \right\|_2 \leq \frac{1}{N} \sum_{i=1}^{N} \left\| \nabla f_i(x_{\tau_{i,k}}) - \nabla f_i(x_k) \right\|_2 + \left\| \nabla f_{i_k}(x_k) - \nabla f_{i_k}(x_{\tau_{i_k,k}}) \right\|_2 + \left\| \nabla f(x_k) - \nabla f(x^*) \right\|_2$$
$$\leq 2LB + 2LB + 2LB = 6LB. \tag{42}$$

The claim is then proved.

$\square$

With the above two claims, we can then bound $V_\tau$ as

$$V_\tau = f(x_\tau) - f(x^*) + L\alpha^2 \sum_{j=1}^{\tau} (\tau - j + 1) \|g_{\tau-j}\|_2^2$$
$$\leq \frac{L}{2} \|x_\tau - x^*\|_2^2 + L\alpha^2 \tau^2 \cdot 36L^2 B^2$$
$$\leq \frac{L}{2} \cdot 4B^2 + 36L^3 \tau^2 B^2 \alpha^2 \tag{43}$$
$$\leq 2LB^2 + \frac{35}{256} LB^2 \leq 3LB^2,$$

where we used smoothness, Claim B.1 and B.2.

## B.2. Modified Scheme

In this case, no updates are made during the first $\tau$ iterations, yielding $x_k = x_0 = 0$ for all $k \leq \tau$. As such, we can readily bound the gradient norm $\|g_k\|_2$ as

$$\|g_k\|_2 \leq 6L \|x_0 - x^*\|_2 = 6L \|x^*\|_2, \quad \forall k \leq \tau, \tag{44}$$

following the proof of Claim B.2 in Appendix B.1. $V_\tau$ is then bounded as

$$
\begin{aligned}
V_\tau &= f(x_\tau) - f(x^*) + L\alpha^2 \sum_{j=1}^{\tau} (\tau - j + 1) \|g_{\tau-j}\|_2^2 \\
&\leq \frac{L}{2} \|x^*\|_2^2 + L\alpha^2 \tau^2 \cdot 36L^2 \|x^*\|_2^2 \\
&\leq 2L \|x^*\|_2^2,
\end{aligned}
\tag{45}
$$

where we use the fact that $\alpha = 1/(16L\tau)$. This demonstrates that $\|x^*\|_2^2$ suffices to capture the burn-in effects if no updates are made during the burn-in period.

## C. Proof of Lemma 3.11

To begin with, we introduce the Markov sampling version of Bernstein's inequality in Theorem 3.4 of Paulin (2015).

**Theorem C.1.** *(Paulin, 2015, Theorem 3.4) [Bernstein's Inequality for Markov Chains] Let $X_1, \cdots, X_k$ be a time-homogeneous, ergodic, and stationary Markov chain $\mathcal{M}$ that takes values in a finite state space $\Omega$. Let $\gamma_{ps}$ and $\pi$ be the pseudo spectral gap and stationary distribution, respectively, of $\mathcal{M}$. Suppose $f$ is a measurable function in $L^2(\pi)$, satisfying $|f(x) - \mathbb{E}_\pi[f]| \leq M, \forall x \in \Omega$, where $\mathbb{E}_\pi$ is the expectation w.r.t. $\pi$. Then, for all $t > 0$, the following is true:*

$$\mathbb{P}\left(S - \mathbb{E}_\pi[S] \leq -t\right) \leq \exp\left(-\frac{t^2 \cdot \gamma_{ps}}{8\left(k + 1/\gamma_{ps}\right)V_f + 20tM}\right), \tag{46}$$

*where $S := \sum_{i=1}^k f(X_i)$, and $V_f := \mathbb{V}_\pi[f]$ is the variance of $f$ under $\pi$.*

According to Proposition 3.4 of Paulin (2015), the pseudo spectral gap can be related to the mixing time $t_{mix}$ defined in Section 3.5 as:

$$\gamma_{ps} \geq \frac{1}{2t_{mix}}. \tag{47}$$

As a result, if $k \geq t_{mix}$, we then have

$$\begin{aligned}
\mathbb{P}\left(S - \mathbb{E}_\pi[S] \leq -t\right) &\leq \exp\left(-\frac{t^2}{16t_{mix}\left(k + 2t_{mix}\right)V_f + 40t_{mix}Mt}\right) \\
&\leq \exp\left(-\frac{t^2}{48kt_{mix}V_f + 40t_{mix}Mt}\right).
\end{aligned} \tag{48}$$

Now we are at a position to apply this inequality to our setting where the sampling indices correspond to the $X_i$'s in Theorem C.1. Fix a component $i \in [N]$ and the starting iteration $k = k_0$. Consider the event that component $i$ is not sampled within any window of length $\tau$ and set

$$f(x) = \mathbb{I}\{x = i\}. \tag{49}$$

Then define $S_{i,\tau,k_0}$ as

$$S_{i,\tau,k_0} = \sum_{k=k_0}^{k_0+\tau-1} f(i_k) = \sum_{k=k_0}^{k_0+\tau-1} \mathbb{I}\{i_k = i\}, \tag{50}$$

which counts visits to component $i$ in a length-$\tau$ window. Since $\mathbb{E}_\pi[S_{i,\tau,k_0}] = \tau\pi_i$, where $\pi_i$ is the entry of component $i$ in $\pi$, the event $\{S_{i,\tau,k_0} \leq 0\}$ is equivalent to

$$S_{i,\tau,k_0} - \mathbb{E}_\pi[S_{i,\tau,k_0}] \leq -\tau\pi_i. \tag{51}$$

On the RHS of (48), we have $M = 1$ since

$$|f(x) - \mathbb{E}_\pi[f]| = \max\{\pi_i, 1 - \pi_i\} \leq 1. \tag{52}$$

We can also compute $V_f$ as

$$V_f = \mathbb{V}_\pi[\mathbb{I}\{x = i\}] = \pi_i(1 - \pi_i) \leq \pi_i. \tag{53}$$

With the specifications above, we can apply (48) as follows:

$$\begin{aligned}
\mathbb{P}\left(S_{i,\tau,k_0} \leq 0\right) &= \mathbb{P}\left(S_{i,\tau,k_0} - \mathbb{E}_\pi[S_{i,\tau,k_0}] \leq -\tau\pi_i\right) \\
&\leq \exp\left(-\frac{\tau^2\pi_i^2}{48\tau t_{mix}\pi_i + 40\tau t_{mix}\pi_i}\right) \\
&= \exp\left(-\frac{\tau\pi_i}{88t_{mix}}\right) \\
&\leq \exp\left(-\frac{\tau\pi_{\min}}{88t_{mix}}\right),
\end{aligned} \tag{54}$$

where in the last step we used the definition of $\pi_{\min}$. Following the same union bound argument in Lemma 3.1, requiring (54) to be smaller than $\delta/(NK)$ and union bounding over all components $i \in [N]$ and iterations $0 \leq k \leq K - 1$ yields the desired claim in Lemma 3.11.

