# OpenReview forum: "A Short and Unified Convergence Analysis of the SAG, SAGA, and IAG Algorithms"
_ICML.cc/2026/Conference — ICML 2026 regular_

### Official Review · Reviewer_bes2 · 2026-03-09

**Soundness:** 2
**Presentation:** 2
**Significance:** 2
**Originality:** 2
**Overall Recommendation:** 4
**Confidence:** 1

**Summary:**

This paper presents a short and unified convergence analysis of  SAG, SAGA, and IAG algorithms. The authors also consider the Markov Sampling and high-probability convergence rate.

**Compliance With Llm Reviewing Policy:**

Affirmed.

**Final Justification:**

Based on my reviewing experience, the score assigned to this paper is very high, which makes me feel that I may not have fully understood the paper. Therefore, I would respectfully suggest that my review not be given any weight in your final decision.

**Key Questions For Authors:**

see Weaknesses

**Strengths And Weaknesses:**

Strengths:

1. The authors present a new and relatively simple framework that unifies several variance-reduction (VR) algorithms and establishes their high-probability convergence rates.

2. The paper also considers other sampling schemes, and the corresponding convergence analysis appears to remain fairly simple and clean.

Weaknesses:

1. Although the proposed proof technique is simple, the framework is still mainly limited to VR algorithms and covers only three specific methods. In addition, VR methods are not directly applicable to many modern machine learning tasks, such as large language model training. More importantly, I do not find the theoretical results to be particularly novel or especially surprising.

2. The proofs for the Markov chain setting seem to contain some issues. Even for Markov gradient descent, the analysis typically requires at least a couple of pages, so I find it difficult to understand why the proof here is so short.

---

> ### Author Rebuttal · Authors · 2026-03-29
>
> Dear Reviewer bes2,
>
> Thank you for your thoughtful review of our paper; we are glad to know that you find our proofs simple and clean.
>
> Below, we respond to each of your main comments. We use 'W' to denote weaknesses.
>
> - **[W1]** We address the following two concerns: (i) the relevance of variance-reduction (VR) methods, and (ii) the novelty of our proof techniques.
>
> We first clarify the importance of VR methods. Since their introduction, methods such as SAG and SAGA have become standard tools for solving *finite-sum optimization* problems, which arise **ubiquitously** in machine learning tasks. This includes classical problems such as *logistic regression, least-squares, support vector machines, and regularized empirical risk minimization*, as detailed in the survey paper [R1] below.
>
> [R1] Gower et al., Variance-Reduced Methods for Machine Learning, Proc. of the IEEE, 2020.
>
> Beyond these settings, VR techniques have been extended to **non-convex optimization, distributed and federated learning (e.g., SCAFFOLD, FedLin), and compositional objectives**. Many communication-efficient federated algorithms incorporate variance-reduction ideas to mitigate client drift and improve convergence.
>
> Finally, beyond supervised learning, we note that several VR methods have been successfully combined with policy gradient algorithms in reinforcement learning (RL) to improve their sample-complexity. See, for instance, the introduction section of [R2] below for a list of such papers.
>
> [R2] Ding et al., On the global optimum convergence of momentum-based policy gradient, AISTATS 2022.
>
> Thus, while we cannot comment on the applicability of VR methods to LLM training specifically, in our humble opinion, this *does not diminish their broad utility across the plethora of ML and RL problems alluded to above*. In this regard, Reviewer isGN mentions: "SAG, SAGA, and IAG are classical and important algorithms, and giving a common conceptual explanation of their convergence behavior is a meaningful and relevant goal."
>
> Also, please see our responses to Reviewers 1J2R and cswP, where we explain that our framework can be extended to cover more general single-loop VR algorithms (beyond SAG and SAGA).
>
> **On Novelty**: Our main contribution is a novel and concise analysis that yields:
>
> 1. **Unified analysis of SAG, SAGA, and IAG.** Despite differences (biased vs. unbiased, stochastic vs. deterministic), our framework captures all three.
>
> 2. **HP guarantees.** Prior results are only in expectation; we provide unified HP bounds capturing tail behavior.
>
> 3. **Improved bounds for IAG.** We *significantly* improve the rate from $\mathcal{O}(\exp(-K/(\kappa^2 N^2)))$ to $\mathcal{O}(\exp(-K/(\kappa N)))$.
>
> 4. **Extension to Markov sampling.** Our analysis applies beyond I.I.D. sampling to Markovian settings.
>
> The fact that a **single** proof framework can achieve all the above features for such classical and extensively studied methods is indeed novel, and non-obvious a priori.
>
> Reviewer isGN considers our approach "elegant and pedagogically valuable", Reviewer cswP commends our "novel angle of analysis", and Reviewer 1J2R mentions that our proofs are "simple, clear, and very well explained."
>
> - **[W2]** With all due respect, we do not believe there is any issue with our Markov sampling proof; if so, we kindly ask the Reviewer to point it out.
>
> We clarify that our Markovian setting differs from the classical *additive Markovian noise* setting where the stochasticity in gradients arises not from sub-sampling, but from generic additive noise in the gradients.  In contrast, in our finite-sum setting, **stochasticity arises from sub-sampling**, and the sampled index sequence follows a Markov chain.
>
> We are able to exploit the above stochastic sub-sampling structure in our two-step proof. First, we establish a high-probability (HP) bounded-staleness event on which each component is visited sufficiently often. In the Markov case, this follows from Bernstein-type inequalities for Markov chains: ergodicity implies that after a burn-in period, samples behave approximately like draws from the stationary distribution of the Markov chain. For ergodic chains, the stationary distribution has positive entries, ensuring sufficient visitation to all component functions, resulting in bounded delays.
>
> Second, conditioned on this event, the analysis boils down to that of delayed gradient descent, and proceeds **identically** to the I.I.D. case, not depending on the sampling process at all. This decoupling explains why the proof remains concise.
>
> Summarily, stochastic sub-sampling, whether I.I.D. or Markov, leads to a bounded delay event (like those in Lemma 3.1 and 3.11). Conditioned on this event, the remainder of the proof is deterministic and agnostic to the sampling scheme.
>
> We hope our rebuttal above has addressed your main comments. If so, we kindly request you to reassess your score.
>
> We would be happy to address any remaining concerns.

---

> > ### Author Rebuttal · Reviewer_bes2 · 2026-04-02
> >
> > I may not fully understand this paper. I have increased the score.

---

> > > ### Author Response · Authors · 2026-04-03
> > >
> > > Thank you for your thoughtful comments, and for raising your score.

---

### Official Review · Reviewer_1J2R · 2026-03-11

**Soundness:** 4
**Presentation:** 4
**Significance:** 3
**Originality:** 3
**Overall Recommendation:** 5
**Confidence:** 4

**Summary:**

This paper studies three algorithms: SAG, SAGA, and IAG. The first two algorithms are classic stochastic variance-reduced algorithms, while IAG is an algorithm with deterministic data ordering. The paper provides high probability rates for these algorithms under IID and Markovian sampling strategies of data points. To the best knowledge, these rates are the first high-probability results for variance-reduced algorithms like SAG and SAGA. The rates almost match the well-known in-expectation results for the algorithms. However, a slightly worse rate comes from the difference between high-probability and in-expectation types of analysis, as explained by the authors, and cannot be avoided. The authors provide the rates under the assumptions: (i) each $f_i$ is $L$-smooth and $f$ is $\mu$-strongly convex, (ii) each $f_i$ is $L$-smooth. The proofs are simple, clear, and very well explained.

**Compliance With Llm Reviewing Policy:**

Affirmed.

**Final Justification:**

I found this work as an elegant theoretical contribution to the field of optimization. The provided proving high-probability framework for studying algorithms with sampling is beautiful and simple, which I believe will motivate more studies for other algorithms. Therefore, I support the acceptance of this work.

**Key Questions For Authors:**

- I found this paper [2] that studies algorithms with different data orderings, and the incremental gradient is a special case of their analysis. Although the work is not directly comparable, it still shares some similarities in the unification of the bounds for various algorithms. I believe it is worth including it in the related works.

- Can other variance-reduced algorithms be covered by this framework? What about SVRG or L-SVRG, SARAH? Can the authors provide more details about the generality of their framework?


[2] Koloskova A, Doikov N, Stich SU, Jaggi M. On convergence of incremental gradient for non-convex smooth functions. arXiv preprint arXiv:2305.19259. 2023 May 30.

**Limitations:**

The paper does not discuss the limitations of this work. In my view, they should add a sentence that states that the framework covers only SAG and SAGA among variance-reduced algorithms.

**Strengths And Weaknesses:**

***Strengths.***

1. Very clean high-probability proofs of convergence for SAG, SAGA, and IAG. The proofs look correct; I did not find mistakes.

2. One of the key steps is showing that the staleness of component gradients can be upper-bounded with high probability. This means that for a sufficiently large window $\tau$, each component $i\in[N]$ will be sampled with high probability.

3. The rates almost match those given by in-expectation analysis. The difference is the following: for the provided high-probability rates, the contraction factor is $\mathcal{O}(1-\frac{1}{\kappa\tau})$, where $\tau=\tilde{\mathcal{O}}(N)$, while for the in-expectation rates it is $\mathcal{O}(1-\min\\{1/\kappa, 1/N\\})$. Here $\kappa$ is the condition number and $N$ is the size of the dataset. The high probability rate is slightly worse by a factor $\tau$. Based on the tight analysis of the delayed gradient descent $x\_{k+1} = x_k - \alpha\nabla f(x\_{k-\tau})$, where $\tau$ is a fixed delay, the authors argue that such a difference in the rates should be expected.

4. The rates further extended to the Marcovian sampling setting. The rates in such a case are affected by the ratio of the mixing time $t_{mix}$ and minimum visitation probability $\pi_{min}$, typically observed in the literature. To the best of my knowledge, there is no analysis that studies variance-reduced algorithms under this sampling assumption.

5. The rates were further extended to the convex case and the IAG method. For the IAG method, the authors improve upon known results in the literature.

6. The construction of the Lyapunov function is clearly explained.

***Weaknesses.***

1. Typically, the bounds in the strongly convex case depend on $\\|x\_0 - x^{\star}\\|$, where $x^{\star}$ is the global solution. The rates in this case depend on $B=\max\\{\\|x^{\star}\\|, \\|x\_1^{\star}\\|, \dots, \\|x\_N^{\star}\\|\\}$, where $x\_i^{\star}=\text{argmin}\_xf\_i(x)$. Having $B$ in the bound is not standard, and should be discussed. My main concern here is that this requires the set of minimizers for each function to be bounded. However, since there is no restriction on each $f_i$ to be convex, the set of minimizers might be unbounded when $f_i$ is non-convex. I encourage the authors to provide a detailed discussion on this aspect. Moreover, it would be good to compare $\\|x\_0 - x^{\star}\\|$ and $B$: which one is larger and when?

2. The authors use $x_0=0$ in the proofs, but don't mention it in the statements. Therefore, the dependency of the rates on the initial conditions $\\|x_0\\|$ or $f(x_0)$ is hidden. Can the authors provide more details for a general case when $x_0\neq0$?

3. It would be good to check the tightness of the bounds on toy problems. For example, the presence of the factor $\tau$ in the rate can be tested by running the algorithms for, say, $100$ random seeds with theoretical values of the stepsize, and then plotting the percentiles for the loss curves (see [1] for an example).

[1] Chezhegov, Savelii, et al. "Clipping improves adam-norm and adagrad-norm when the noise is heavy-tailed." arXiv preprint arXiv:2406.04443 (2024).

---

> ### Author Rebuttal · Authors · 2026-03-29
>
> Dear Reviewer 1J2R,
>
> Thank you for your thoughtful review of our paper and your encouraging comments; we are glad to know that you find our proofs simple, clear, and very well explained.
>
> Below, we respond to each of your main comments. We use 'W' to denote weaknesses and 'Q' to denote questions.
>
> - **[W1]** Indeed, the pre-factor $B$ is worse than classical in-expectation constants such as $\lVert x_0-x^*\rVert ^2$. In what follows, we clarify its origin and explain how one can recover classical bounds using a simple and intuitive algorithmic modification.
>
> Our choice of Lyapunov function $V_k$ is well-defined only for $k \geq \tau$. Thus, during the burn-in period $\tau$, we cannot say much about progress towards the optimum $x^*$. Moreover, during this period, all components may not have been visited even once. As such, when $k < \tau$, updates rely on incomplete gradient information and can be biased toward local minimizers $x_i^\star$. This leads to a uniform bound $\lVert x_k\rVert\le B$ (Lemma 3.8), which in turn yields the larger pre-factor.
>
> This dependence can be removed via a simple modification: **freeze updates for the first $\tau$ iterations**, using this phase only to populate gradient memory. Then $x_k=x_0$ for $k\le\tau$, and one can bound
> $$
> \lVert g_k\rVert \le 6L\lVert x_0-x^\star\rVert, \quad k\le\tau.
> $$
> This gives
> $$
> V_\tau \le 2L\lVert x_0-x^\star \rVert^2,
> $$
> and the final bound becomes
> $$
> f(x_K)-f(x^*) \le 4L\lVert x_0-x^\star \rVert^2 (1-1/(64\tau\kappa))^K,
> $$
> matching classical pre-factors.
>
> We will add the aforementioned simple burn-in modification to our revised paper.
>
> - **[W2]** We thank the Reviewer for pointing this issue out. We will add $x_0=0$ in the revised version.
>
> For the general case with $x_0 \neq 0$, redefine $B$ as
> $$
> B := \max(\lVert x^* \rVert, \lVert x_1^* \rVert,\cdots,\lVert x_N^* \rVert,\lVert x_0 \rVert).
> $$
> The proof then proceeds identically as in Appendix B. Moreover, under the modification to SAG/SAGA in our response to W1, we obtain $B=\lVert x_0-x^*\rVert$, which matches classical bounds.
>
> - **[W3]** We thank the Reviewer for this excellent suggestion, and we plan to include such simulations in our revised paper to test the tightness of our exponent w.r.t. $\tau.$
>
> - **[Q1]** We thank the Reviewer for pointing us to the nice paper [2] which focuses on in-expectation results for the non-convex setting. In contrast, we mainly focus on high-probability bounds for the strongly convex, smooth setting. We will cite and discuss this paper in our revised version.
>
> - **[Q2]** We thank the Reviewer for this question. In short, our framework extends naturally to single-loop VR methods (e.g., SAG, SAGA, IAG) and their variants (proximal, accelerated, second-order, etc.,), while extension to double-loop methods (e.g., SVRG, SARAH) is possible but may not yield additional insights.
>
> For **single-loop algorithms** with stochastic sub-sampling, at each iteration, only one component gradient is evaluated, and the iterates evolve continuously without periodic re-initializations, allowing us to track the algorithm along a single trajectory.
>
> Exploiting this structure, we can continue to establish a 'bounded delay' event (Lemma 3.1), where the staleness of all component gradients is bounded. Importantly, for single-loop methods, staleness is inherently **random**, which motivates the bounded-delay construction. Conditioned on this event, we conjecture that the rest of the analysis should follow similarly for other VR methods, where we first establish a one-step descent lemma (Lemma 3.2), and bound the gradient error (Lemma 3.3). The latter would then inform the choice of the Lyapunov function for the specific VR method under study.
>
> In contrast, double-loop methods such as SVRG periodically reset their reference points and gradient estimators. In this case, the staleness is deterministically bounded by the outer-loop length, eliminating the need for a HP event. While a similar Lyapunov analysis may still be possible, it is unclear whether new insights would emerge from it.
>
> **Summary.** Our framework is particularly suited to single-loop methods with stochastic sub-sampling, where staleness is random and must be controlled probabilistically. The same proof strategy can extend to related methods that preserve this structure, with appropriate modifications to the Lyapunov function. We will add a detailed discussion.
>
> **Response to Limitations.** We thank the Reviewer for this helpful suggestion. As discussed in our response to Q2, we will add a discussion on the generality of our framework, explaining that it is not limited to just SAG and SAGA, but can also be conceptually extended to a broader class of single-loop methods (e.g., second-order, constrained, accelerated, distributed, etc.) with stochastic sub-sampling with suitable modifications.
>
> We hope our rebuttal above has addressed your main comments. We would be happy to address any remaining concerns.

---

> > ### Author Rebuttal · Reviewer_1J2R · 2026-04-02
> >
> > Thank you for providing clarifications! I don't have further questions.
> >
> > I encourage the authors to incorporate the discussion about $B$ and single-loop VR algorithms into the paper to improve clarity and comparison to prior works. I am happy to support the acceptance of this work, which provides an important theoretical framework for studying finite-sum problems via high-probability techniques. I am curious to see the applications of the theoretical framework beyond VR methods (maybe it's worth mentioning in the conclusion as well).

---

> > > ### Author Response · Authors · 2026-04-03
> > >
> > > Thank you once again for your valuable comments; we appreciate them!
> > >
> > > In the revised version, we will incorporate all your suggestions.

---

### Official Review · Reviewer_cswP · 2026-03-13

**Soundness:** 4
**Presentation:** 4
**Significance:** 3
**Originality:** 3
**Overall Recommendation:** 5
**Confidence:** 4

**Summary:**

The authors present a novel analysis of well-known stochastic gradient algorithms, namely, SAG, SAGA, and IAG, whose available proofs of convergence are different, typically long, and not easily interpretable. By viewing these algorithms as, essentially, a deterministic gradient descent algorithm operating with stale information about the gradients of the component functions of the total loss, the authors not only achieve a unified proof, but also one that is easy to interpret and can be extended seamlessly to non-convex objectives and markovian sampling  of the component functions. The proof is rather short and elegant, which is a fresh and welcomed exception in an otherwise mainstream of sprawling proofs presented with little insight.

**Compliance With Llm Reviewing Policy:**

Affirmed.

**Final Justification:**

The authors have addressed my concerns in their rebuttal. Therefore I maintain my positive score of 5.

**Key Questions For Authors:**

Q1. Can this approach to analyzing convergence be applied to other well-known stochastic gradient methods such as SVRG (Johnson & Zhang, 2013)?

Minor suggestion: the use of the Bernstein's inequality in (10) is not needed; as the random variables $Y_{k,i}$  follow a Bernoulli distribution, we have directly $$P\left(\sum_{k = k_0}^{k_0 + \tau -1} Y_{i,k} \leq 0 \right) = (1 - p)^\tau = ( 1 - 1/N )^\tau \leq e^{-\tau/N},$$ the last inequality following from $1 - x \leq e^{-x}$ for all $x$. This bound is slightly tighter than the derived bound of $e^{-3 \tau / (8 N)}$.

Typos:
- line 120, second column, "initialized to \tau_{i,k} = 0 for all i,k": I think you mean "for all i and k = 0";
- line 720, ".. and N denotes the cardinality": I believe it should be "M denotes the cardinality."

**Limitations:**

Yes.

**Strengths And Weaknesses:**

Strengths:
- Novel angle of analysis that unifies the convergence analysis of standard stochastic-gradient algorithms (SAG, SAGA, and IAG);
- Proof is simple to understand (e.g., the new Lyapunov function is crafted in an intuitive manner);
- The presentation of the paper is outstanding, the authors guiding the reader step-by-step and caring to highlight the main takeaways after each major theorem.

Weaknesses:
- The analysis is only in the form of high-probability bounds, while previous proofs report on the expected performance.

---

> ### Author Rebuttal · Authors · 2026-03-29
>
> Dear Reviewer cswP,
>
> Thank you for your thoughtful review of our paper and your encouraging comments; we are glad to know that you find our proof elegant and our presentation outstanding.
>
> Below, we respond to each of your main comments. We use 'W' to denote weaknesses and 'Q' to denote questions.
>
> - **[W1]** Indeed, our current analysis only provides high-probability (HP) bounds. Before explaining why, let us note that a key motivation of our work is precisely the lack of HP analyses for VR methods such as SAG and SAGA, as prior work focuses almost exclusively on in-expectation guarantees. Our contribution establishes and unifies HP analyses for SAG/SAGA, while also improving rates for IAG.
>
> We now clarify why our technique does not directly yield in-expectation bounds. Theorem 3.9 states that with $\tau=\tilde{\mathcal{O}}(N\log(NK/\delta))$ and $\alpha=1/(16L\tau)$, with probability at least $1-\delta$, for $K>\tau$,
> $$
> f(x_K)-f(x^*) \le 6LB^2\left(1-\frac{1}{64\tau\kappa}\right)^K.
> $$
>
> On the "bad" event where the above bound does not hold, suppose for simplicity that the iterates remain bounded (e.g., by projection), i.e., $\lVert x_k\rVert\le C$. Then
> $$
> \mathbb{E}[f(x_K)-f(x^*)] \le 6LB^2\exp\left(-\frac{K}{64\tau\kappa}\right) + \mathcal{O}(\delta C^2).
> $$
>
> To obtain exponential decay, one would set $\delta=\mathcal{O}(\exp(-K))$, making $\delta C^2$ negligible. However, this choice yields $\tau=\tilde{\mathcal{O}}(NK)$, causing the final bound to be
> $$
> \mathbb{E}[f(x_K)-f(x^*)] \le \mathcal{O}(1) + \mathcal{O}(\exp(-K)),
> $$
> which is vacuous. Thus, there is an inherent trade-off: making $\delta$ small suppresses the failure term but enlarges $\tau$, weakening contraction. At the moment, we do not have a way to overcome this issue. We will explain this point in the revised paper.
>
> - **[Q1]** In short, our framework extends naturally to single-loop VR methods (e.g., SAG, SAGA, IAG) and their variants (proximal, accelerated, second-order, etc.), while extension to double-loop methods (e.g., SVRG, SARAH) is possible but may not yield additional insights.
>
> For **single-loop algorithms** with stochastic sub-sampling, at each iteration, only one component gradient is evaluated, and the iterates evolve continuously without periodic re-initializations, allowing us to track the algorithm along a single trajectory.
>
> Exploiting this structure, we can continue to establish a ''bounded delay'' event (Lemma 3.1), where the staleness of all component gradients is bounded. Importantly, for single-loop methods, staleness is inherently **random**, which motivates the bounded-delay construction. Conditioned on this event, we conjecture that the rest of the analysis should follow similarly for other VR methods, where we first establish a one-step descent (Lemma 3.2), and bound the gradient error (Lemma 3.3). The latter would then inform the choice of the Lyapunov function for the specific VR method under study.
>
> In contrast, double-loop methods such as SVRG periodically reset their reference points and gradient estimators. In this case, the staleness is **deterministically bounded** by the outer-loop length, eliminating the need for a HP event. While a similar Lyapunov analysis may still be possible, it is unclear whether new insights would emerge from it.
>
> **Summary.** Our framework is particularly suited to single-loop methods with stochastic sub-sampling, where staleness is random and must be controlled probabilistically. The same proof strategy can extend to related methods that preserve this structure, with appropriate modifications to the Lyapunov function.
>
> - **[Q2]** We thank the Reviewer for this valuable suggestion. Indeed, under I.I.D. uniform sampling, directly exploiting the Bernoulli structure of $Y_{i,k}$ yields a slightly sharper bound for the staleness parameter $\tau$.
>
> Let us justify our use of Bernstein’s inequality. Our analysis in Section 3.5 extends beyond I.I.D. sampling to the Markovian setting, where indices are generated by a time-homogeneous Markov chain and are temporally correlated. In this case, the Bernoulli-based argument may no longer apply, and establishing the desired HP staleness bound requires concentration inequalities that account for such correlations.
>
> To this end, we employ a variant of Bernstein’s inequality for Markov processes, which allows us to handle both I.I.D. and Markovian sampling within a unified framework. We will clarify this point in our revised paper, and add a comment mentioning that the Bernoulli argument can lead to sharper bounds for the I.I.D. case.
>
> - **[Q3]** We sincerely thank the Reviewer for their careful inspection. We will fix the first typo. For the second, $N$ does denote the cardinality of the state space of $\mathcal{M}$ (see also Section 3.5), and $M$ is the constant in $|f(x)-\mathbb{E}_\pi[f]|\leq M$.
>
> We hope our rebuttal above has addressed your main comments. We would be happy to address any remaining concerns.

---

> > ### Author Rebuttal · Reviewer_cswP · 2026-04-02
> >
> > I thank the authors for addressing the questions I raised and maintain my positive score. Congratulations to the authors for the work.

---

> > > ### Author Response · Authors · 2026-04-03
> > >
> > > Thank you once again for your valuable comments; we appreciate them!

---

### Official Review · Reviewer_isGN · 2026-03-14

**Soundness:** 3
**Presentation:** 3
**Significance:** 3
**Originality:** 3
**Overall Recommendation:** 5
**Confidence:** 5

**Summary:**

This paper studies convergence of the SAG, SAGA, and IAG algorithms for finite-sum optimization. The main contribution is a short and unified proof framework based on viewing these methods through a bounded-delay perspective. For SAG and SAGA, the paper derives **high-probability** linear convergence guarantees in the smooth strongly convex setting, and also discusses extensions beyond this basic regime. For IAG, the same proof strategy yields a deterministic linear rate and appears to significantly improve the best previously known dependence on the condition number and the problem size.

**Compliance With Llm Reviewing Policy:**

Affirmed.

**Final Justification:**

In my opinion this paper is a solid accept.

**Key Questions For Authors:**

A positive feature is that the dependence on failure probability is only logarithmic, through $\log(1/\delta)$. However, $\delta$ enters through the contraction rate by way of $\tau$, rather than appearing as a separate deviation term or prefactor. This is somewhat unusual and makes the final guarantee less interpretable than more standard high-probability bounds. I do not view this as a fatal issue, and it may well be inherent to the bounded-delay proof strategy, but it is worth noting.

Do you think it is possible to obtain a dependence outside the exponential term instead?

**Strengths And Weaknesses:**

**Strengths**
- SAG, SAGA, and IAG are classical and important algorithms, and giving a common conceptual explanation of their convergence behavior is a meaningful and relevant goal. The paper articulates this motivation well and makes a convincing case that existing analyses are quite fragmented, especially between SAG/SAGA and IAG.

- The bounded-delay event is easy to interpret, and once this event is established, the Lyapunov analysis becomes modular and conceptually simple. I found this perspective elegant and pedagogically valuable. Even if not all resulting rates are optimal (see weaknesses), the proof technique itself is a real contribution.

- Beyond the unified analysis, the authors derive stronger high-probability bounds, featuring only a logarithmic depedence on the failure probability, compared to in-expectation analysis. Furthermore, the rate achieved for IAG improves the dependence on key parameters compared to SOTA.

**Weaknesses**
- For the strongly convex SAG/SAGA result, after replacing the value of $\tau$ in the bound one obtains (roughly)

$f(x_K) - f(x^*) \leq 6LB^2 \exp(- K / (C \kappa N \log(NK/\delta))).$

Thus the effective exponent scales like $1 / (\kappa N \log(NK/\delta))$, which is substantially weaker than the classical in-expectation rates for SAG/SAGA. This does not invalidate the contribution, but it does limit the strength of the result on the randomized side.

- Even if the authors made a discussion about $B$ this parameter may by substantially worse than the initialization constant appearing in the classical in-expectation results, and imho is a bit unnatural.

- The proof fixes a target horizon $K$ and chooses $\tau$ accordingly via a union bound over the sampling process up to that horizon. As a result, the theorem is not of the form: for all $k \geq 0$ simultaneously with probability at least $1-\delta$. This is weaker than the usual infinite-horizon in-expectation statements one has for SAG/SAGA. Again, this is understandable given the proof technique, but it is a limitation of the final result.

---

> ### Author Rebuttal · Authors · 2026-03-29
>
> Dear Reviewer isGN,
>
> Thank you for your thoughtful review of our paper and your encouraging comments; we are glad to know that you find our proof idea elegant and pedagogically valuable.
>
> Below, we respond to each of your main comments. We use 'W' to denote weaknesses and 'Q' to denote questions.
>
> - **[W1 +Q1]**  It is  true that our exponent depends on $1/(\kappa N)$, which is worse than the in-expectation rate of SAG/SAGA that depends on $\min(1/\kappa,1/N)$. However, to our knowledge, no prior work establishes high-probability (HP) guarantees for these methods, nor corresponding lower bounds, so the tightness of our exponent remains unclear. We conjecture that the $\kappa N$ dependence cannot be improved too much.
>
> **First**, GD satisfies
> $$
> f(x_K)-f(x^\star)\le (1-1/\kappa)^K(\cdot).
> $$
> Lemma 3.1 shows that w.h.p., all components are visited within $\tau=\tilde{\mathcal{O}}(N)$ iterations. Thus, $\tau$ SAG/SAGA steps correspond to one full gradient pass, yielding a slowdown by $N$. It seems unlikely that the concentration bound in Lemma 3.1 can be tightened significantly.
>
> **Second**, this discrepancy stems from the role of expectation. In SAGA [1], the Lyapunov function decomposes as $T_k = R_k + c Q_k$, where $R_k$ (table error) and $Q_k$ (iterate error) enter the recursion with *separate coefficients* after taking expectation. These correspond to conditioning ($1/\kappa$) and table refresh ($1/N$), leading to the rate $\min(1/\kappa,1/N)$.
>
> In contrast, our HP analysis does not admit such decoupling. The iterate and table terms are *deterministically coupled* via Facts 3.5 and 3.6 in our paper, and without taking expectations we cannot average out randomness. Thus, contraction and staleness must be controlled *simultaneously* along a single trajectory, yielding a multiplicative dependence $1/(\kappa N)$.
>
> **Finally**, this is consistent with delayed GD results [2], which show a rate $\exp(-K/(\tau\kappa))$ is *tight*. Since our analysis reduces to a deterministic delayed system under bounded staleness, this further suggests that the additional $\tau=\tilde{\mathcal{O}}(N\log(NK/\delta))$ term is intrinsic.
>
> **Response to [Q1]**: We agree that while $\log(1/ \delta)$ terms typically enter HP bounds as a pre-factor, this is typically true for scenarios where the convergence rate is polynomially (and not exponentially) decaying in $K$, such as Theorem 3.10 of our paper. When exponential in $K$ rates are possible, based on our discussion above, the slow-down by a factor of $\tau$ seems reasonable, and $\tau$ scales with $\log(1/\delta)$, explaining the source of the latter in our exponent.
>
> [1] Defazio et al., SAGA: A fast incremental gradient method with support for non-strongly convex composite objectives. NIPS, 2014.
>
> [2] Arjevani et al., A tight convergence analysis for stochastic gradient descent with delayed updates, ALT 2020.
>
> - **[W2]** We thank the Reviewer for this insightful comment. Indeed, the pre-factor $B$ is worse than classical in-expectation constants such as $\lVert x_0-x^*\rVert ^2$. In what follows, we clarify its origin and explain how one can recover classical bounds using a simple and intuitive algorithmic modification.
>
> Our choice of Lyapunov function $V_k$ is well-defined only for $k \geq \tau$. Thus, during the burn-in period $\tau$, we cannot say much about progress towards the optimum $x^*$. Moreover, during this period, all components may not have been visited even once. As such, when $k < \tau$, updates rely on incomplete gradient information and can be biased toward local minimizers $x_i^\star$. This leads to a uniform bound $\lVert x_k\rVert\le B$ (Lemma 3.8), which in turn yields the larger pre-factor.
>
> This dependence can be removed via a simple modification: **freeze updates for the first $\tau$ iterations**, using this phase only to populate gradient memory. Then $x_k=x_0$ for $k\le\tau$, and one can bound
> $$
> \lVert g_k\rVert \le 6L\lVert x_0-x^\star\rVert, \quad k\le\tau.
> $$
> This gives
> $$
> V_\tau \le 2L\lVert x_0-x^\star \rVert^2,
> $$
> and the final bound becomes
> $$
> f(x_K)-f(x^*) \le 4L\lVert x_0-x^\star \rVert^2 (1-1/(64\tau\kappa))^K,
> $$
> matching classical pre-factors.
>
> We will add the aforementioned simple burn-in modification to our revised paper.
>
> - **[W3]** To remove prior knowledge of the horizon $K$,  we conjecture that one can use the standard "doubling trick". We run SAG/SAGA in epochs (indexed by $j$'s) of length $K_j=2^j$, setting the delay $\tau_j$ for epoch $j$ based on $K_j$. Let $J$ be such that $2^{J+2}>K$, then the last epoch has length $2^J \ge K/4$, so we lose out by a factor of at most $4$ (relative to when $K$ is known). A union bound over epochs (with suitable failure probabilities for each epoch) ensures the guarantees hold simultaneously for all $k$, without knowing $K$. We will elaborate on this in the revised paper.
>
> We hope our rebuttal above has addressed your main comments. We would be happy to address any remaining concerns.

---

> > ### Author Rebuttal · Reviewer_isGN · 2026-04-01
> >
> > I thank the authors for their rebuttal which addressed almost all my questions in full. I believe this is a good paper with a solid and valuable contribution for which I recommend acceptance.

---

> > > ### Author Response · Authors · 2026-04-03
> > >
> > > Thank you once again for your insightful and positive comments. We appreciate it.

---

### Decision · Program_Chairs · 2026-04-30

**Decision:**

Accept (regular)

**Comment:**

This paper studies variance-reduction algorithms, namely SAG, SAGA, and IAG. Relative to standard analyses, the approach developed here is original and relies on controlling the delay/staleness in the component gradient updates of the finite-sum problem. This control is obtained through a careful use of Bernstein's inequality.

Building on these elements, the authors develop a unified Lyapunov analysis for SAG, SAGA, and IAG. I found the SAG and IAG parts particularly compelling, especially the SAG analysis, which has historically been rather intricate. More broadly, the paper succeeds in providing a short, clean, and interpretable proof framework for these methods.

The reviewers also identified some limitations, in particular that the randomized results are stated in high probability rather than in expectation, which leads to somewhat weaker rates than the classical ones, and that the overall scope remains limited to a relatively small family of variance-reduction methods. Nevertheless, the consensus is that the proof technique is elegant, the presentation is very strong, and the paper makes a meaningful contribution to our understanding of this class of methods.

Overall, I find the contribution solid. The reviewers unanimously recommend acceptance, and I concur.